photochemistry/materials science

concentration gradient, cell culture, photodegradable hydrogel, microfluidic device, micropatterning

**Author for correspondence:**
Shinji Sugiura
e-mail: shinji.sugiura@aist.go.jp

[†]Present address: Center for Medical Sciences, St Luke's International University, 3-6-2 Tuskiji, Chuo-ku, Tokyo, 104-0045, Japan.
This article has been edited by the Royal Society of Chemistry, including the commissioning, peer review process and editorial aspects up to the point of acceptance.

# Stepwise construction of dynamic microscale concentration gradients around hydrogel-encapsulated cells in a microfluidic perfusion culture device

Shinya Yamahira[1,†], Taku Satoh[1], Fumiki Yanagawa[1], Masato Tamura[1], Toshiyuki Takagi[1], Eri Nakatani[1,2], Yuta Kusama[1,2], Kimio Sumaru[1], Shinji Sugiura[1] and Toshiyuki Kanamori[1]

[1]Biotechnology Research Institute for Drug Discovery, National Institute of Advanced Industrial Science and Technology (AIST), Central 5, 1-1-1 Higashi, Tsukuba 305-8565, Japan
[2]Department of Bioengineering, Nagaoka University of Technology, 1603-1 Kamitomioka, Nagaoka, Niigata 940-2188, Japan

SS, 0000-0002-1070-2482

Inside living organisms, concentration gradients dynamically change over time as biological processes progress. Therefore, methods to construct dynamic microscale concentration gradients in a spatially controlled manner are needed to provide more realistic research environments. Here, we report a novel method for the construction of dynamic microscale concentration gradients in a stepwise manner around cells in micropatterned hydrogel. In our method, cells are encapsulated in a photodegradable hydrogel formed inside a microfluidic perfusion culture device, and perfusion microchannels are then fabricated in the hydrogel by micropatterned photodegradation. The cells in the micropatterned hydrogel can then be cultured by perfusing culture medium through the fabricated microchannels. By using this method, we demonstrate the simultaneous construction of two dynamic concentration gradients, which allowed us to expose the cells encapsulated in the hydrogel to a dynamic microenvironment.

# 1. Introduction

Many biological processes rely on concentration gradients that dynamically change over time as the processes progress for processes including embryo development [1], immune responses [2], wound healing [3] and angiogenesis [4]. Furthermore, the spatial patterns of the concentration gradients involved often vary over time. For example, the concentration gradients of signalling molecules and the timing of signals regulate spatial pattern formation of body in embryogenesis [5,6]. Therefore, the ability to construct dynamic concentration gradients is needed to investigate such biological events.

Recently, methods for creating concentration gradients of ions [7], small molecules [8], peptides [9], proteins [10], physiological parameters of the extracellular matrix [11,12] and cells [13] have been developed to construct research environments that mimic those found *in vivo*. To generate these gradients, diffusion [14], heat [15], gravity [16], electric force [17], light [18] and precise fluid control using microfluidic devices [12,19–22] have been used. However, in most of these previous studies, the source of the biological compound was fixed in position and the concentration gradient remained constant over time with a predetermined spatial pattern. There have been only limited studies to report on generation of concentration gradients of multiple compounds using microfluidic devices [19,20]. In these systems, the spatial pattern of concentration gradients is defined by the structure of the microchannel. A versatile method capable of changing the spatial pattern of concentration gradients potentially creates a new opportunity to recapitulate dynamic biological events.

Compared with the other technologies currently available for generating concentration gradients, the use of photoresponsive hydrogels has many advantages [23]. The photolithographic process provides fabrication of high-resolution spatial pattern at the sub-micrometre scale [24], and it can be applied anywhere in the transmissive space, which allows for the construction of concentration gradients in a stepwise manner. Indeed, other groups have reported that cellular functions can be controlled using a polyethylene glycol (PEG) hydrogel with a three-dimensional stiffness gradient generated by two-photon lithography [25]. Cell differentiation can be controlled by patterning multiple compounds in a hydrogel via a stepwise photochemical reaction [26]. Furthermore, hydrogel-based culture can provide a more *in vivo*-like environment for cells and artificial tissues compared with two-dimensional culture [27,28]. Thus, photolithographic fabrication of hydrogels could be an advantageous approach for constructing concentration gradients for examining dynamic biological events.

Here, we developed a method for constructing concentration gradients with a designed spatial pattern using micropatterned degradation of a photodegradable hydrogel that was formed inside of a microfluidic perfusion culture device. To produce the photodegradable hydrogel, we used dibenzocyclooctyl-terminated photocleavable tetra-arm-polyethylene glycol (DBCO-PC-4armPEG), which is a click cross-linkable and photocleavable PEG that we developed in our previous study [29], and azide-terminated tetra-arm-PEG (azide-4armPEG) (figure 1). The hydrogel was formed inside the microfluidic device via a simple two-component mixing reaction, and micropatterned photodegradation was used to fabricate perfusion microchannels and 'diffusion reservoirs' in the hydrogel. Finally, multiple concentration gradients were constructed by the stepwise micropatterned photodegradation and natural swelling of the hydrogel to control the flow of fluids into the different diffusion reservoirs.

# 2. Material and methods

## 2.1. Synthesis of dibenzocyclooctyl-terminated photocleavable tetra-arm-polyethylene glycol

*N*-Hydroxysuccinimide (NHS)-terminated tetra-arm-PEG (NHS-4armPEG; 4-armPEG-succinimidyl carboxymethyl ester, 10 k; molecular weight (MW): 10 258) was obtained from JenKem Technology USA (Plano, TX). Using the NHS-4armPEG, we synthesized DBCO-PC-4armPEG, as reported previously [29]. As a result, 7.3 g of DBCO-PC-4armPEG (MW: 13 013) was obtained at a yield of 98.6%. The structure was confirmed by proton nuclear magnetic resonance in deuterated chloroform; the spectrum showed a peak from the DBCO group at approximately 7.3 ppm (electronic supplementary material, figure S1 in the electronic supplementary information).

## 2.2. Synthesis of azide-terminated tetra-arm-polyethylene glycol

Azide-4armPEG was synthesized from amino-terminated 4armPEG (SUNBRIGHT PTE-100PA, MW: 9617; NOF Co., Tokyo, Japan) using azide-PEG4-NHS ester (Click Chemistry Tools LLC, Scottsdale,

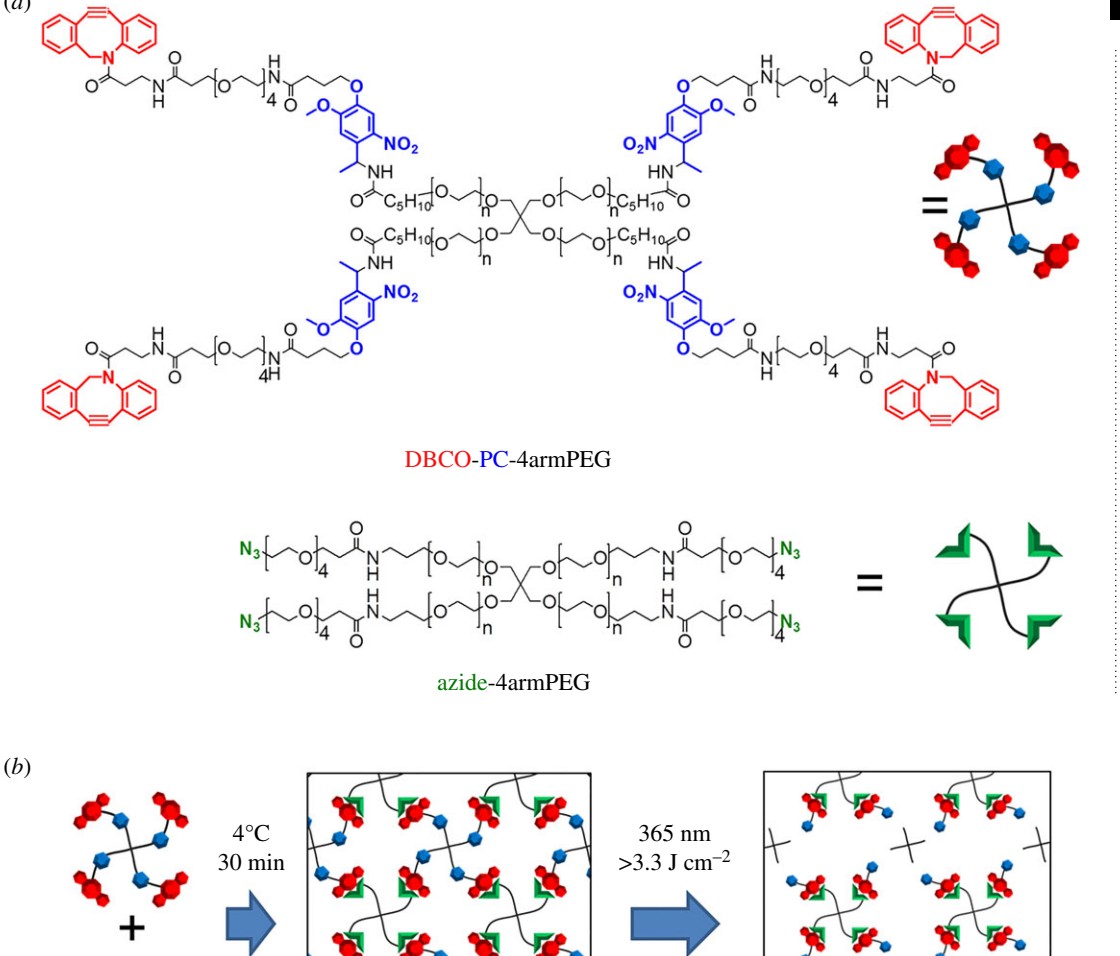

**Figure 1.** Schematic showing an overview of the study set-up. (*a*) Molecular structures of DBCO-PC-4armPEG and azide-4armPEG. (*b*) Formation of the photodegradable hydrogel via the reaction between DBCO-PC-4armPEG and azide-4armPEG, and photodegradation of the photodegradable hydrogel.

AZ), as reported previously [29]. Amino-terminated 4armPEG (50 mmol, 4.9 ml) and azide-PEG4-NHS ester (200 mmol, 0.1 ml) were dissolved in 300 mM HEPES buffer (pH 7.4). To promote the reaction between the amino-base and the NHS-activated ester group, the mixture was incubated in a water bath at 37°C for 2 h. The mixture was then transferred to a dialysis tube (MW cut-off: 6–8 k; Spectrum laboratories, Inc., Rancho Dominguez, CA) and dialyzed against 5 l of Milli-Q water (Millipore, Billerica, USA) for 24 h. The outer water was refreshed at 30 min and 1.5, 3 and 18 h. The dialyzed mixture was freeze-dried using a freeze dryer (FDS-1000; Tokyo Rikakikai, Co. Ltd, Tokyo, Japan). As a result, 508 mg of azide-4armPEG (MW: 10 710) was obtained at a yield of 91%. The molar modification rate of the synthesized azide-4armPEG was estimated as 98.5% by fluorescamine assay [30].

## 2.3. Fabrication of the microfluidic perfusion culture device

The microfluidic perfusion culture device was designed by incorporating the pressure-driven medium circulation mechanism we reported previously (figure 2*a*) [31,32]. Briefly, medium circulation was generated by applying sequential pressure to the feed and storage reservoirs. Medium flow to the storage reservoir through the culture channel was generated by applying pressure to the feed reservoir while flow through the return channel was prohibited by the Laplace valve and elevated aperture of the glass pipe in the feed reservoir. Medium flow to the feed reservoir through the return channel

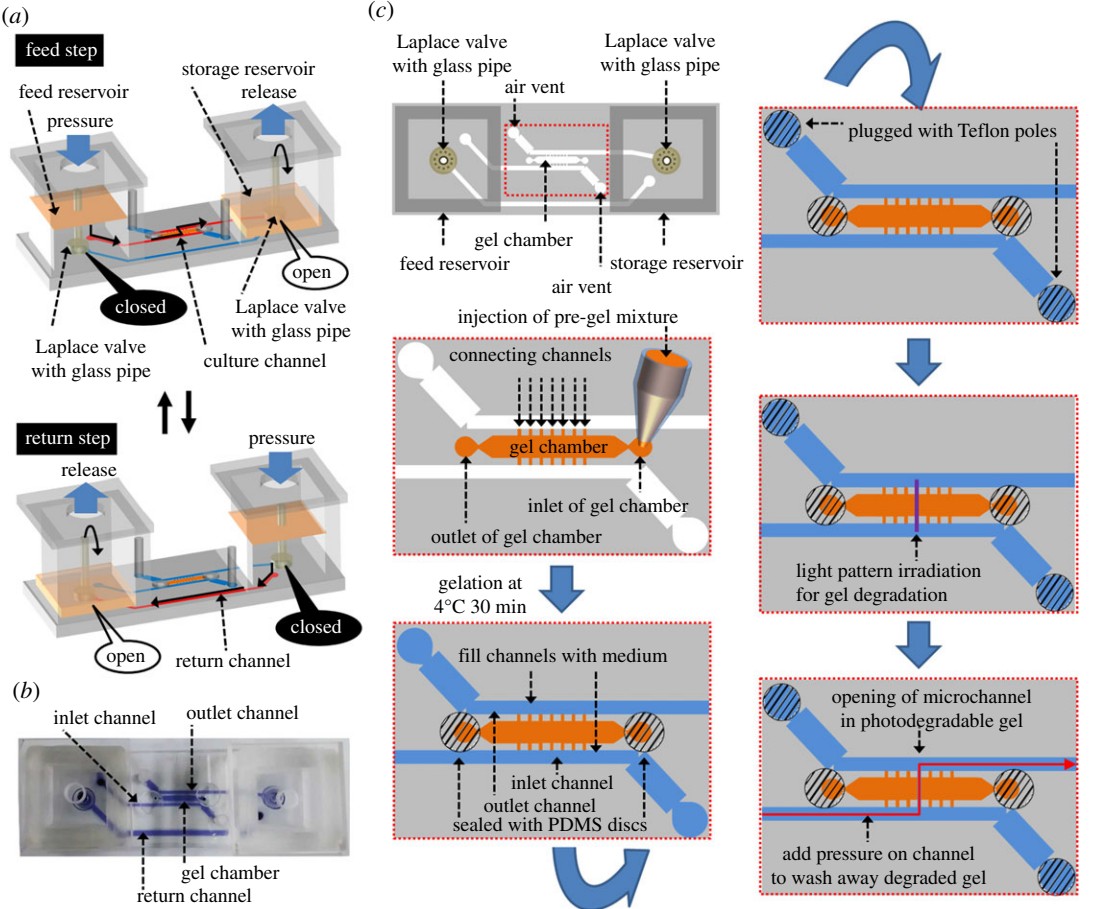

**Figure 2.** The microfluidic perfusion culture device. (*a*) Semi-one-way medium circulation using two Laplace valves and sequential application of pneumatic pressure. (*b*) Photograph of the fabricated microfluidic perfusion culture device. (*c*) Preparation of the photodegradable hydrogel inside the microfluidic perfusion culture device, and the process for fabricating a microchannel in the photodegradable hydrogel. PDMS, polydimethylsiloxane.

was generated by applying pressure to the storage reservoir while flow through the culture channel was prohibited by the same mechanism described above.

The microfluidic device was fabricated out of polydimethylsiloxane (PDMS, silpot184 silicon elastomer, Toray Dow Corning, Tokyo, Japan) by soft lithography from a master mould made out of the negative photoresist SU-8 (MicroChem, MA) [33,34]. Photomasks for three-layer photolithography were designed in Adobe Illustrator CS4 (Adobe, Inc., San Jose, CA). The first layer was for the Laplace valves (final depth: $26 \pm 3\,\mu m$) and was fabricated using SU-8 3025 on a silicon wafer. The second layer was for the connecting channels (final depth: $118 \pm 3\,\mu m$) and was fabricated using SU-8 2050 on top of the first layer. The third layer was for the construction of the gel chamber and microchannels (final depth: $228 \pm 4\,\mu m$) and was fabricated using SU-8 2075 on top of the second layer. After developing in ethyl lactate, the master mould was treated with fluorinated trichloro(1H,1H,2H,2H-perfluorooctyl)silane (Sigma-Aldrich, St Louis, MO). A 10 : 1 mixture of PDMS prepolymer and catalyst was poured into the master mould and baked at 80°C for 2 h. The demoulded PDMS chip was punched to create holes for the culture medium inlet and outlet, return channels and an air vent, and holes for hydrogel filling, and then bonded to a glass slide by oxygen plasma treatment. Glass pipes for pressure-driven circulation of culture medium were attached to the holes surrounded by the Laplace valves. Feed and storage reservoirs were also bonded to the microfluidic device by oxygen plasma treatment. The fabricated microfluidic device is shown in figure 2*b*.

## 2.4. Formation of the photodegradable hydrogel inside the microfluidic device

A photodegradable hydrogel was formed in the gel chamber (width: 1.5 mm; length: 8 mm) in the microfluidic device via the click reaction in which DBCO moieties in DBCO-PC-4armPEG and azide moieties in azide-4armPEG react spontaneously to form covalent cross-links (figure 2*c*). Solutions (7 mM

each) of azide-4armPEG and DBCO-PC-4armPEG were prepared in Dulbecco's modified Eagle's medium (DMEM, low glucose; Thermo Fisher Scientific, MA) supplemented with 10% fetal bovine serum (FBS) and mixed together at a 1 : 1 volume ratio. Next, 5 µl of the mixture was introduced into the gel chamber via the inlet of the gel chamber, while the mixed solution did not flow out to inlet and outlet channels from thin connecting channels of the microfluidic device due to capillary action of the hydrophobic PDMS surface. After the inlet and outlet holes were covered with PDMS discs (figure 1c, left-bottom panel), the click reaction for gelation was conducted at 4°C for 30 min. After the inlet and outlet channels were filled with culture medium, the air vents were plugged with Teflon poles (figure 1c, right-top panel), and then the microfluidic device was assembled with an aluminium holder, glass lid with female Luer port and acrylic holder (electronic supplementary material, figure S2).

## 2.5. Micropatterned photodegradation of the hydrogel and perfusion culture

Micropatterned photodegradation of the hydrogel was performed with a micropattern projection system (DESM-01; Engineering System, Co. Ltd, Matsumoto, Japan). A bitmap image file (1024 × 768 pixel) of light pattern was designed, and the patterned light as designed was irradiated on the hydrogel for degradation (theoretical resolution on the focal plane: 3.1 µm pixel$^{-1}$; wavelength: 365 nm; 72.5 mW cm$^{-2}$) (figure 1c, right-middle panel) [29,35]. To open the microchannels, pneumatic pressure was applied through the Luer filter using a diaphragm pump (EAP-01, AS ONE, Osaka, Japan) with a pressure regulator (PR-4102, GL Science, Tokyo, Japan) and a manometer (PG-100–102 GP, Copal Electronics, Tokyo, Japan), and then medium was perfused through the device (figure 1c, right-bottom panel). The upper pressure limit to break the Laplace valve was measured as 5.2 kPa ± 0.1 (n = 3), so pressures less than this limit were used during perfusion. For continuous perfusion, sequential pressure was applied to the feed and storage reservoirs by using the diaphragm pump, pressure regulator and a pneumatic pressure control system (ASTF0401, Engineering System) [31,32], and semi-one-way medium circulation was generated by the Laplace valves and elevated apertures of the glass pipes (figure 2a). The details of the semi-one-way circulation mechanism are described in our previous reports [31,32].

## 2.6. Cell culture and encapsulation in the photodegradable hydrogel

Human hepatocellular carcinoma cells (HepG2) were obtained from RIKEN (Tsukuba, Japan) and cultured in DMEM containing 10% FBS and 1% penicillin-streptomycin. The cells were suspended in DBCO-PC-4armPEG solution and embedded into the photodegradable hydrogel at $1.78 \times 10^7$ cells ml$^{-1}$ by mixing with azide-4armPEG solution.

## 2.7. Microscopy and image analysis

Fluorescence and phase-contrast microscope images were acquired with an inverted microscope (IX71; Olympus, Tokyo, Japan). The acquired images were processed with the ImageJ software (1.48v, NIH).

# 3. Results

## 3.1. Fabrication of microchannels in the photodegradable hydrogel

First, we evaluated the minimum width of microchannel that could be produced in the photodegradable hydrogel. After formation of the hydrogel in the gel chamber of the device, the outlet channel was filled with DMEM (10% FBS), and the inlet channel was filled with 0.5 µM fluorescein isothiocyanate (FITC)–dextran (MW: 40 k; Sigma-Aldrich, MO) in DMEM (10% FBS) to allow us to visualize the microchannels. Lines of widths from 6.25 to 50 µm (figure 3a, top image) were produced by irradiating the hydrogel with ultraviolet light between the inlet and outlet channels for 2 min. Then, pressure was applied to the feed reservoir at 5 kPa for 10 min, and the hydrogel was observed under a fluorescence microscope (figure 3a). We found that the fluorescent solution did not flow through the 6.25 µm and 12.5 µm microchannels, indicating that the microchannel was not fully formed. However, the fluorescent solution did flow through the 25 µm and 50 µm microchannels (figure 3a, bottom image, and electronic supplementary material, figure S3). Thus, we found the photodegradation condition to form microchannel width as small as 25 µm. We also found microchannels wider than 50 µm were preferable for flow through under the condition we tested.

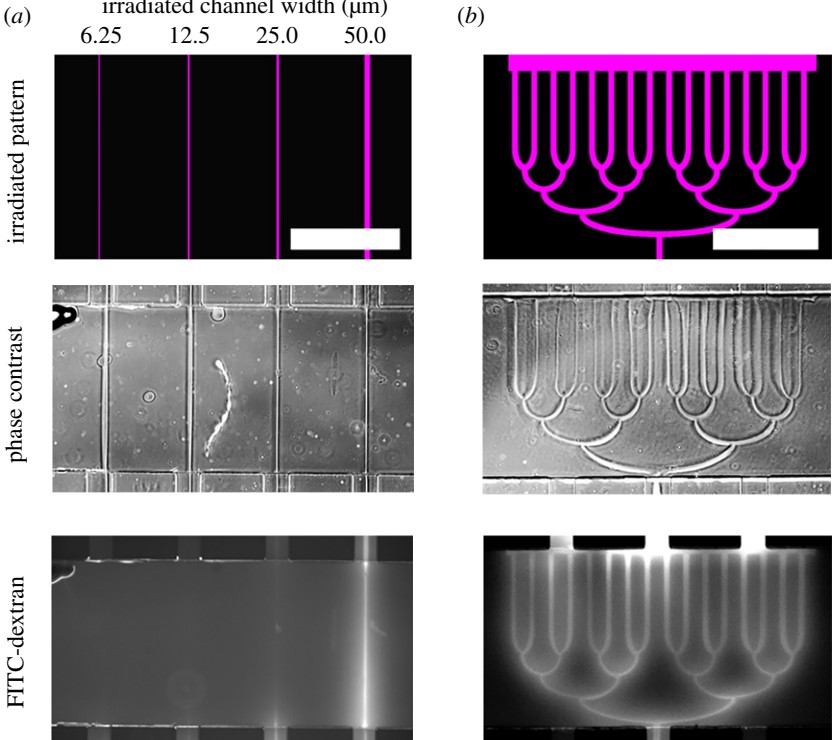

**Figure 3.** Fabrication of microchannels in photodegradable hydrogel by micropatterned light irradiation. Pressure was applied from the bottom in these images. (*a*) Examination of the minimum channel width for microchannel fabrication. (*b*) A 1-to-16 branching microchannel network was fabricated in the hydrogel. Scale bars: 1 mm.

Next, we examined the fabrication of a complex microchannel structure in the hydrogel. This time, a 1-to-16 branching microchannel network with a constant channel width of 50 μm (figure 3*b*, top image) was produced by irradiating the hydrogel for 45 s, and pressure was applied to the feed reservoir at 3 kPa for 10 min to open the channels; all of the channels were confirmed to be open (figure 3*b*).

## 3.2. Diffusion behaviour of fluorescently labelled molecules in the photodegradable hydrogel

Various hydrogels have been developed as scaffolds for cell culture, and the diffusion behaviours of biological molecules in these hydrogels have been evaluated in many studies [36]. Elucidation of the diffusion behaviour of biological molecules is important because the supply of nutrients and dynamics of metabolism during cell growth are important factors for successful three-dimensional cell culture in hydrogels. To examine the effect of MW on diffusion behaviour in the photodegradable hydrogel, we fabricated four 200 μm wide microchannels in the hydrogel spaced 600, 1400 and 2200 μm apart (figure 4). We then perfused the hydrophilic fluorescent dye calcein (MW: 0.4 k; Sigma-Aldrich, MO) or FITC–dextran (MW: 4 k or 40 k) into the microchannels at 5 kPa of applied pressure and examined the diffusion of the compounds at 0, 15, 30, 60, 120 and 240 min. As expected, the speed of diffusion increased with decreasing MW; a particularly obvious difference in diffusion speed was observed between FITC–dextran 4 k (figure 4*b*) and 40 k (figure 4*c*).

## 3.3. Micropatterned perfusion culture of HepG2 cells encapsulated in the photodegradable hydrogel

Next, we examined the effect of hydrogel compartment size on cell viability during perfusion culture. We have confirmed cell viability in the off-device hydrogel as a preliminary examination (electronic supplementary material, figure S4). Most of the encapsulated cells were alive after 3 days of culture in the 350 μm thick photodegradable hydrogel. After encapsulation of HepG2 cells in the hydrogel, four perfusion microchannels each with a width of 200 μm were photofabricated to construct three hydrogel compartments with widths of 600, 1400 and 2200 μm, as described above. After 3 days of

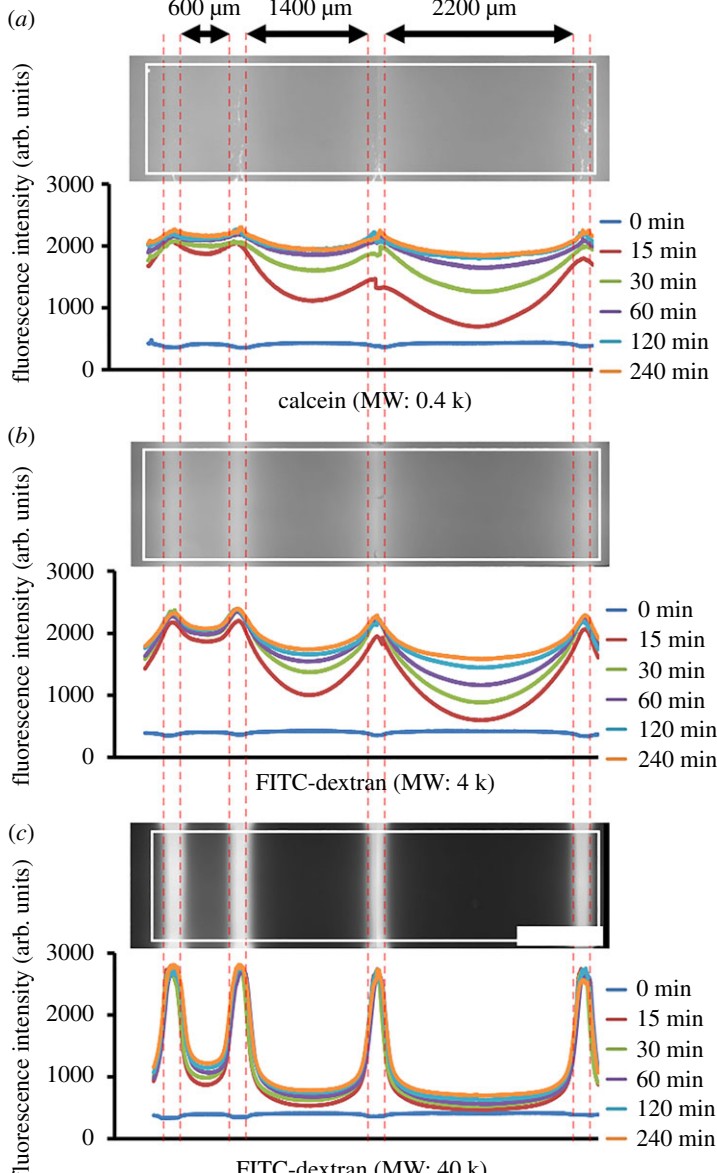

**Figure 4.** Effect of molecular weight on molecular diffusion in the photodegradable hydrogel. Red dashed lines in each image indicate the bounds of the irradiated microchannels. White solid square lines indicate the areas of fluorescence intensity analysed, which are shown below each image. Arrows above the top panel represent the area where the three hydrogel compartments remained after photofabrication of four 200 µm-wide microchannels. Scale bar: 1 mm.

perfusion culture with 1 min feed and 4 min return steps and 5 kPa of pneumatic pressure, dead cells were stained with 20 µM ethidium homodimer-1 (Life Technologies, Carlsbad, CA) in DMEM (10% FBS) for 6 h. Fluorescent microscopy observation revealed that the ratio of the dead cells in the 600 µm wide compartment was smaller than that in the 1400 µm and 2200 µm wide compartments (figure 5). This indicated that the supply of nutrients to the cells and the removal of wastes were limited to within 300 µm from the perfusion microchannel. We have also tried staining of living cells. However, the cells only in the area close to the perfusion microchannel were stained probably due to the limitation in diffusion of dye and metabolism of dye by living cells (data not shown).

## 3.4. Spatial patterning of multiple concentration gradients

Next, we examined the spatial patterning of two concentration gradients in the hydrogel. A major difficulty associated with the practical use of hydrogels in tissue engineering is preventing swelling of the hydrogel and maintaining its shape [37]. However, here, we harnessed the swelling of the

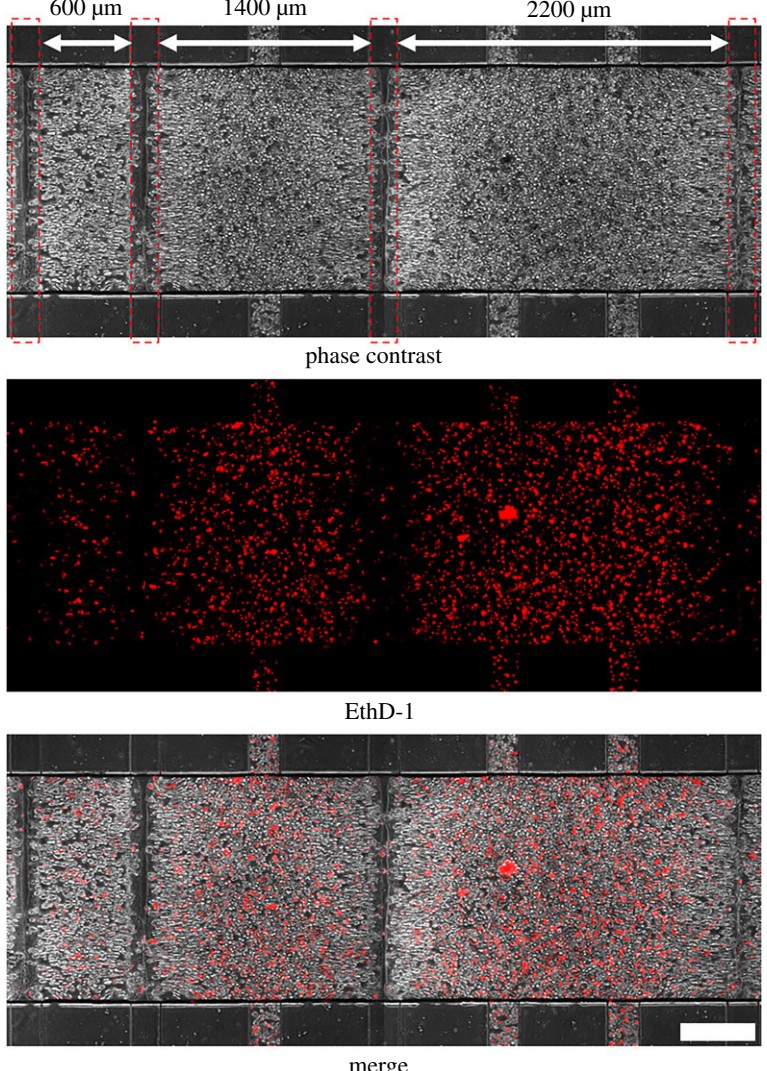

600 µm    1400 µm    2200 µm

phase contrast

EthD-1

merge

**Figure 5.** Hydrogel-embedded HepG2 cells were cultured under continuous perfusion for 3 days. Then, dead cells were stained red with ethidium homodimer-1 (EthD-1). Red dashed lines in the phase-contrast image indicate the bounds of the irradiated microchannels. Scale bar: 0.5 mm.

hydrogel to control the flow of fluids through the microchannels fabricated in the hydrogel; that is, we were able to use the natural expansion of the hydrogel to trap solutions in diffusion reservoirs created in the hydrogel by micropatterned degradation.

Once the photodegradable hydrogel was formed in the gel chamber, the outlet channel was filled with DMEM (10% FBS), and the inlet channel and feed reservoir were filled with 1 µM FITC–dextran (MW: 40 k) in DMEM (10% FBS). We then fabricated the first microchannel, which comprised a fusiform diffusion reservoir and capillary microchannels (see electronic supplementary material, figure S5 for the micropattern), and introduced the FITC–dextran solution into the microchannel by applying 5 kPa of pressure to the feed reservoir for 5 min. After standing at atmospheric pressure at 4°C for 5 min to accelerate swelling, the hydrogel had swelled, closing the microchannel and segregating the FITC–dextran solution in the diffusion reservoir from the flow in the inlet and outlet channels (figure 6, left). After replacing the solution in the feed reservoir and the inlet channel with 1 µM Texas Red–dextran (MW: 40 k, Sigma) in DMEM (10% FBS) at room temperature, we then fabricated a second microchannel, which had a bilaterally symmetrical pattern with respect to the first microchannel, approximately 1.6 mm apart from the first microchannel. The Texas Red–dextran solution was introduced into the second microchannel under the same conditions used for the FITC–dextran solution; the Texas Red–dextran solution flowed into only the second diffusion reservoir because the first microchannel had closed (figure 6, centre, electronic supplementary material, figure S6). After standing at atmospheric pressure at 4°C for 5 min to allow the capillary microchannel in the second microchannel to close, the

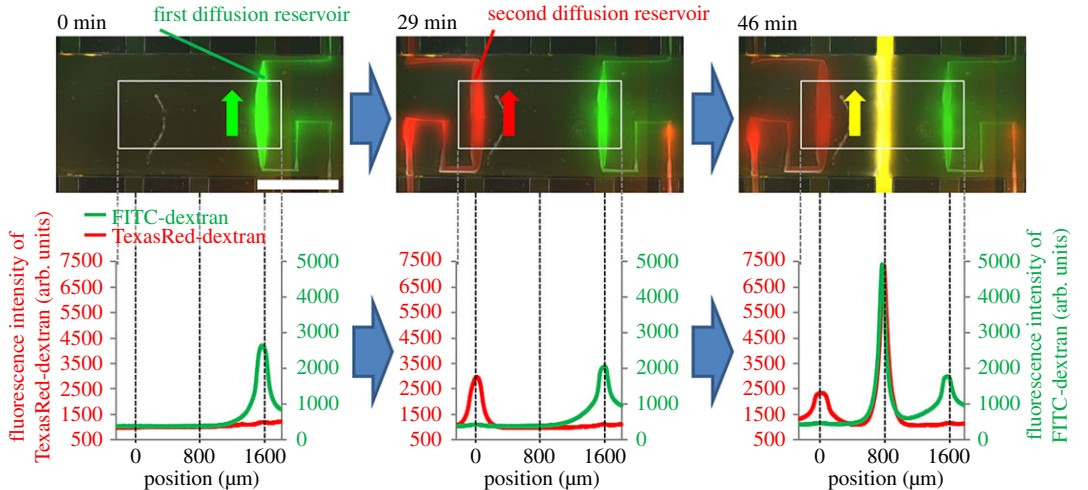

**Figure 6.** Micropatterning of multiple concentration gradients in the photodegradable hydrogel. Upper images are merges of FITC–dextran (green), Texas Red–dextran (red), and phase-contrast images. White solid lines indicate the areas of fluorescent intensity that were analysed; the fluorescence intensity profile is shown below each image. Arrows in the photographs indicate the direction of flow. Scale bar: 1 mm.

solution in the feed reservoir and inlet channel was replaced with a mixture of FITC–dextran and Texas Red–dextran in DMEM (10% FBS) (1 µM each). A straight microchannel with a width of 200 µm was irradiated in the space between the first and second microchannels, and the mixed solution of Texas Red–dextran and FITC–dextran was introduced into the microchannel, as already described. Concentration gradients between the two diffusion reservoirs and the straight microchannel were observed in the hydrogel by fluorescence microscopy (figure 6, right).

## 3.5. Micropatterning of two simultaneous dynamic concentration gradients

Finally, we examined the generation of two simultaneous dynamic concentration gradients within a complicated shape by stepwise degradation of the HepG2 cell-embedded hydrogel. First, a round diffusion reservoir with 450 µm diameter was created and filled with 0.5 µM FITC–dextran (MW: 40 k) in DMEM (10% FBS), as already described. The FITC–dextran solution diffused from the reservoir, and a concentration gradient was formed (figure 7a). About 15 min later, the solution in the inlet channel was changed to Texas Red–dextran (MW: 40 k) in DMEM (10% FBS) solution, and the HepG2 cell-embedded hydrogel was fabricated into a hand shape (figure 7b). The hand shape was designed so that all of the cells were located 300 µm or less from the flow path or diffusion reservoir (figure 5, electronic supplementary material, figure S7). Next, Texas Red–dextran solution was perfused at 5 kPa pressure for 5 min, which caused the Texas Red–dextran solution to diffuse into the hydrogel from the fingertips of the hand. During perfusion, the FITC–dextran solution in the round diffusion reservoir diffused further from the palm to the fingertips of the hand. After perfusion for an additional hour, the Texas Red–dextran gradient was a gentle slope from the fingertip to the palm, and the FITC–dextran gradient had disappeared (figure 7c).

## 4. Discussion

Here, we demonstrated the stepwise fabrication of simple or multi-branched microchannels in a photodegradable hydrogel formed inside a microfluidic perfusion culture device (irradiation time less than 2 min), and the use of this device to construct two simultaneous dynamic concentration gradients.

In characterizing our hydrogel, we found that at 15 min after the start of perfusion, the concentration of calcein (MW: 0.4 k) and a low-MW dextran (MW: 4 k) at the centre of a 600 µm wide hydrogel compartment was comparable to that of the introduced solution (figure 4), whereas the concentration of a high-MW dextran (MW: 40 k) was not. The difference in the diffusion behaviours of the two dextrans was probably a result of the difference in the diffusion coefficients of the two polymers in water and of steric hindrance by the polymer chain, of which the average length of PEG between the

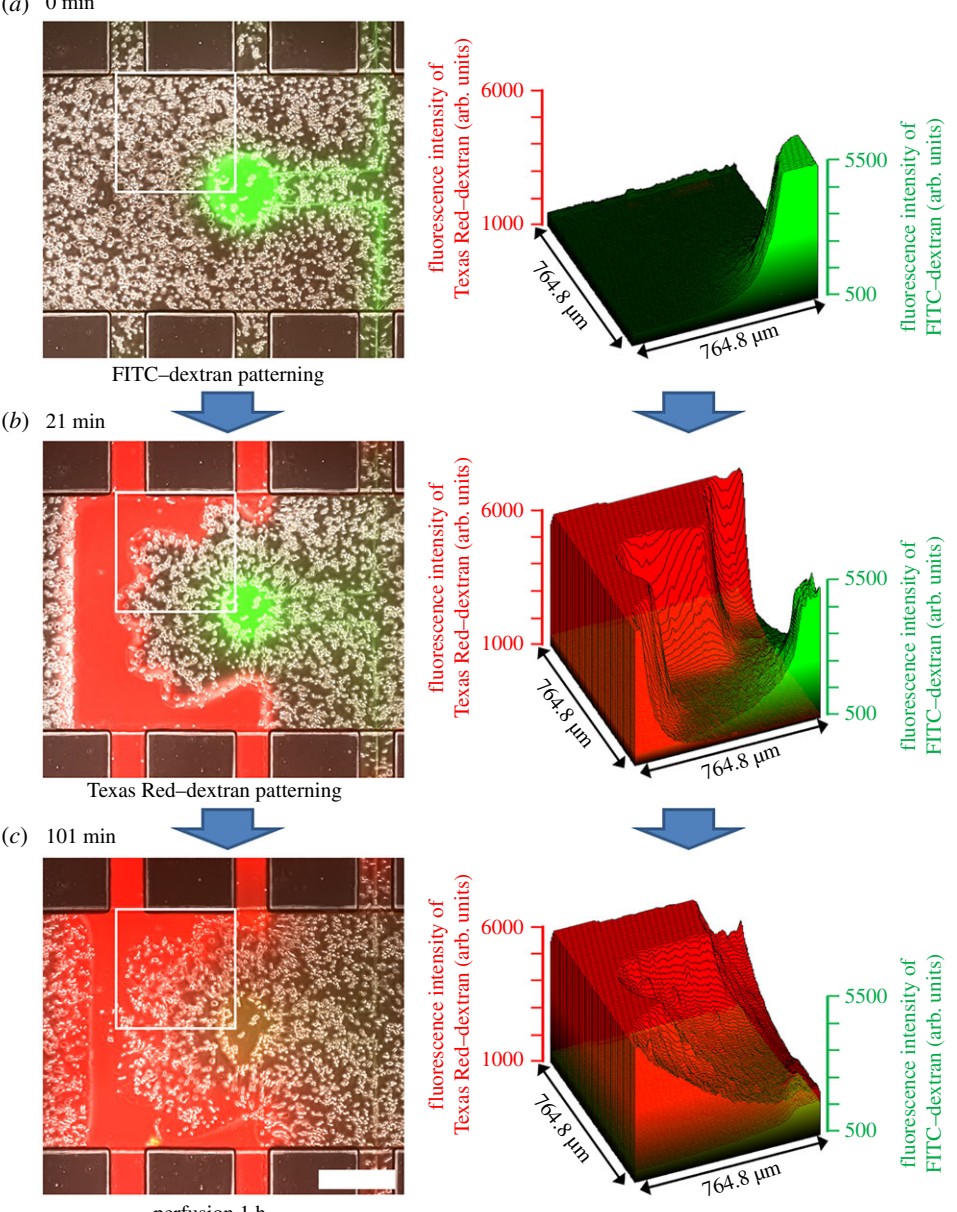

**Figure 7.** Micropatterning of a dynamic two-compound concentration gradient. Left, merges of FITC–dextran (green), Texas Red–dextran (red), and phase-contrast images. White solid lines indicate the areas of fluorescence intensity analysed, which are shown to the right of each image. Scale bar: 500 µm.

network junctions corresponds to approximately 5 k of MW. Also, diffusion of calcein was so fast that a concentration profile of calcein at 15 min indicates the diffusion of calcein during the image acquisition, which takes a couple of minutes without perfusion. Furthermore, when we examined the perfusion culture of HepG2 cells embedded in the hydrogel, the ratio of dead cells in the hydrogel compartment with a width of 600 µm was smaller than that in the wider compartments (figure 5), which indicated that the exchange of nutrients and waste was sufficient to maintain only cells within 300 µm of the perfusion microchannel. This property of the hydrogel is comparable with the situation within the human body, where most living tumours are located within 250 µm of a blood vessel [27].

We then used the photofabrication process to construct two simultaneous concentration gradients. By leveraging the natural swelling of the hydrogel, separated diffusion reservoirs for different compounds were fabricated in the hydrogel (figure 6) without the need for mechanical valves or a complex microfluidic network, as reported previously [20,21]. Because the swelling of hydrogel is ubiquitous, this means of controlling fluid flow is probably applicable to almost any hydrogel.

Many other groups have examined the importance of concentration gradients in various biological processes [4,20–22,38]. However, in most of this research, the compounds and their gradients were limited to the substrate surface or a hydrogel network. By contrast, our system enabled the construction of two simultaneous dynamic concentration gradients during perfusion cell culture (figure 7). Also, there have been a couple of studies reporting the method to generate concentration gradients of multiple compounds using microfluidic devices [19,20]. In these methods, the shape of hydrogel and spatial pattern of the gradients were defined by the structure of microchannels. By contrast, our method used micropatterned degradation of hydrogel and enabled us to change spatial pattern of cell-encapsulated hydrogel in a stepwise manner. In other words, we can change the shape of hydrogel and generate concentration gradients with different spatial pattern in the same-structured microfluidic device.

# 5. Conclusion

In this study, we developed a method for constructing multiple concentration gradients during the perfusion culture of cells. With our system, we were able to fabricate simple and complex microchannels and diffusion reservoirs in a photodegradable hydrogel, which we then used to construct dynamic concentration gradients during perfusion cell culture.

Ethics. This article does not present research with ethical considerations.

Data accessibility. Electronic supplementary material available: NMR spectrum of DBCO-PC-4armPEG (electronic supplementary material, figure S1), schematic showing assembly of the perfusion device (electronic supplementary material, figure S2), heat map of fluorescence intensity (electronic supplementary material, figure S3), cell viability in the off-device hydrogel (electronic supplementary material, figure S4), irradiation patterns used to construct the concentration gradients (electronic supplementary material, figure S5 and S7) and simultaneous micropatterning of two concentration gradients (electronic supplementary material, figure S6).

Authors' contributions. S.S. proposed the concept. M.T., F.Y. and T.T. synthesized the materials. F.Y., T.S. and S.S. designed and S.Y. fabricated the microfluidic devices. S.Y., F.Y., T.S., E.N., Y.K., K.S. and S.S. designed the experiment. S.Y. acquired and analysed the data. S.S. and T.K. supervised the study. S.Y., M.T, and S.S. wrote the manuscript. All authors reviewed and provided comments on the manuscript.

Competing interests. We have no competing interests.

Funding. This study was partially supported by JSPS KAKENHI with grant nos. JP17K19018 and 19H02530.

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
