## [Reviewer comments · Royal Society Open Science]

Review History

RSOS-200027.R0 (Original submission)

Review form: Reviewer 1

Is the manuscript scientifically sound in its present form?

Yes

Are the interpretations and conclusions justified by the results?

Yes

Is the language acceptable?

Yes

Do you have any ethical concerns with this paper?

No

Have you any concerns about statistical analyses in this paper?

Yes

Recommendation?

Accept as is

Comments to the Author(s)

In the current manuscript, Sugiura and coworkers reported a method for the construction of dynamic microscale concentration gradients by using photodegradable hydrogel formed in microfluidic system. They also demonstrate that such kind of system can be in cell culture by perfusing culture medium. This method is very easy and under control. I suggest it can be published as is.

Review form: Reviewer 2**Is the manuscript scientifically sound in its present form?**

No

Are the interpretations and conclusions justified by the results?

No

Is the language acceptable?

Yes

Do you have any ethical concerns with this paper?

No

Have you any concerns about statistical analyses in this paper?

No

Recommendation?

Major revision is needed (please make suggestions in comments)

Comments to the Author(s)

This work reported a method of constructing chemical concentration gradients in photodegradable hydrogels for microfluidic cell culture. The major concerns from the reviewer are as follows:

1. The authors repeatedly emphasized the importance and the construction of "dynamic concentration gradients". However, in the reviewer's opinion, the term "dynamic" used here may be inapposite.
2. The cytocompatibility of both the hydrogel material and the fabrication process should be characterized.
3. The authors should quantitatively characterize cell viability after perfusion cell culture.
4. HepG2 cells were embedded in hydrogels when micropatterning two simultaneous concentration gradients, however, no cell biology results were given.
5. The authors should demonstrate the importance of concentration gradient generated here for cell biology (not just cell survival).

Review form: Reviewer 3 (Stoyan Karakashev)**Is the manuscript scientifically sound in its present form?**

Yes

Are the interpretations and conclusions justified by the results?

Yes

Is the language acceptable?

Yes

Do you have any ethical concerns with this paper?

No

Have you any concerns about statistical analyses in this paper?

No

Recommendation?

Accept as is

Comments to the Author(s)

The authors conducted well designed investigation on the effect of the concentration gradient on the diffusive flux on encapsulated cells in micro-patterned hydrogel. In fact they succeeded to mimic the conditions in a real tissue. Of course this is not a scientific breakthrough as far as the first Fick's law is well known, but the contribution of the paper consists in building up of scaffold in which a real cells can be encapsulated and perfused by real patterned micro-channel. This opens a gate for many more detailed investigations of cultures of cells. I really kike the work and recommend its publication.

Decision letter (RSOS-200027.R0)

03-Apr-2020

Dear Dr Sugiura:

Title: Stepwise construction of dynamic microscale concentration gradients around hydrogel-encapsulated cells in a microfluidic perfusion culture device
Manuscript ID: RSOS-200027

The editor assigned to your manuscript has now received comments from reviewers. We would like you to revise your paper in accordance with the referee and Subject Editor suggestions which can be found below (not including confidential reports to the Editor). Please note this decision does not guarantee eventual acceptance.

Please submit your revised paper before 26-Apr-2020. Please note that the revision deadline will expire at 00.00am on this date. If we do not hear from you within this time then it will be assumed that the paper has been withdrawn. In exceptional circumstances, extensions may be possible if agreed with the Editorial Office in advance. We do not allow multiple rounds of revision so we urge you to make every effort to fully address all of the comments at this stage. If deemed necessary by the Editors, your manuscript will be sent back to one or more of the original reviewers for assessment. If the original reviewers are not available we may invite new reviewers.

To revise your manuscript, log into <http://mc.manuscriptcentral.com/rsos> and enter your Author Centre, where you will find your manuscript title listed under "Manuscripts with Decisions." Under "Actions," click on "Create a Revision." Your manuscript number has been

appended to denote a revision. Revise your manuscript and upload a new version through your Author Centre.

RSC Associate Editor:
Comments to the Author:
(There are no comments.)

RSC Subject Editor:
Comments to the Author:
(There are no comments.)

Reviewers' Comments to Author:
Reviewer: 1

Comments to the Author(s)
In the current manuscript, Sugiura and coworkers reported a method for the construction of dynamic microscale concentration gradients by using photodegradable hydrogel formed in microfluidic system. They also demonstrate that such kind of system can be in cell culture by perfusing culture medium. This method is very easy and under control. I suggest it can be published as is.

Reviewer: 2

Comments to the Author(s)
This work reported a method of constructing chemical concentration gradients in photodegradable hydrogels for microfluidic cell culture. The major concerns from the reviewer are as follows:

1. The authors repeatedly emphasized the importance and the construction of "dynamic

concentration gradients". However, in the reviewer's opinion, the term "dynamic" used here may be inapposite.

2. The cytocompatibility of both the hydrogel material and the fabrication process should be characterized.
3. The authors should quantitatively characterize cell viability after perfusion cell culture.
4. HepG2 cells were embedded in hydrogels when micropatterning two simultaneous concentration gradients, however, no cell biology results were given.
5. The authors should demonstrate the importance of concentration gradient generated here for cell biology (not just cell survival).

Reviewer: 3

Comments to the Author(s)

The authors conducted well designed investigation on the effect of the concentration gradient on the diffusive flux on encapsulated cells in micro-patterned hydrogel. In fact they succeeded to mimic the conditions in a real tissue. Of course this is not a scientific breakthrough as far as the first Fick's law is well known, but the contribution of the paper consists in building up of scaffold in which a real cells can be encapsulated and perfused by real patterned micro-channel. This opens a gate for many more detailed investigations of cultures of cells. I really like the work and recommend its publication.

Author's Response to Decision Letter for (RSOS-200027.R0)

See Appendix A.

RSOS-200027.R1 (Revision)

Review form: Reviewer 2

Is the manuscript scientifically sound in its present form?

Yes

Are the interpretations and conclusions justified by the results?

Yes

Is the language acceptable?

Yes

Do you have any ethical concerns with this paper?

No

Have you any concerns about statistical analyses in this paper?

No

Recommendation?

Accept as is

Comments to the Author(s)

The authors have addressed the comments from this reviewer

Decision letter (RSOS-200027.R1)

Dear Dr Sugiura:

Title: Stepwise construction of dynamic microscale concentration gradients around hydrogel-encapsulated cells in a microfluidic perfusion culture device

Manuscript ID: RSOS-200027.R1

It is a pleasure to accept your manuscript in its current form for publication in Royal Society Open Science. The chemistry content of Royal Society Open Science is published in collaboration with the Royal Society of Chemistry.

RSC Associate Editor:
Comments to the Author:
(There are no comments.)

RSC Subject Editor:
Comments to the Author:
(There are no comments.)

Reviewer(s)' Comments to Author:
Reviewer: 2

Comments to the Author(s)
The authors have addressed the comments from this reviewer

Appendix A

Referee 1

Referee's Comments:

In the current manuscript, Sugiura and coworkers reported a method for the construction of dynamic microscale concentration gradients by using photodegradable hydrogel formed in microfluidic system. They also demonstrate that such kind of system can be in cell culture by perfusing culture medium. This method is very easy and under control. I suggest it can be published as is.

Our response:

We thank reviewer's careful reading, understanding on our work, and evaluation for publication. We revised manuscript in accordance with the other referee's comments. We hope our manuscript is acceptable for publication in Royal Society Open Science.

Referee: 2

Referee's Comments:

This work reported a method of constructing chemical concentration gradients in photodegradable hydrogels for microfluidic cell culture. The major concerns from the reviewer are as follows:

1. The authors repeatedly emphasized the importance and the construction of "dynamic concentration gradients". However, in the reviewer's opinion, the term "dynamic" used here may be inapposite.
2. The cytocompatibility of both the hydrogel material and the fabrication process should be characterized.
3. The authors should quantitatively characterize cell viability after perfusion cell culture.
4. HepG2 cells were embedded in hydrogels when micropatterning two simultaneous concentration gradients, however, no cell biology results were given.
5. The authors should demonstrate the importance of concentration gradient generated here for cell biology (not just cell survival).

Our response:

We thank reviewer's careful reading and valuable comments to make manuscript better. We revised manuscript in accordance with the referee's comments. The revised parts were indicated by the red color in the manuscript and electronic supplementary material. We hope our manuscript is acceptable for publication in Royal Society Open Science.

Answer to the comment 1:

We used the word "dynamic" to express the phenomena that "change over time". In order to clarify this point we added the words "over time" in the first appearance of the word "dynamically" in the summary and introduction section. Actually, we achieved stepwise change of concentration gradient over time. We think that this stepwise change is "dynamic change" because the change took place over time.

Summary:

Inside living organisms, concentration gradients dynamically change **over time** as biological processes progress.

Introduction:

Many biological processes rely on concentration gradients that dynamically change **over time** as the processes progress for processes including embryo development.

Answer to the comment 2:

We have added new experimental data on the cytocompatibility of the hydrogel as Fig. S4. The characterization of cytocompatibility of each fabrication process is difficult because analysis of cytocompatibility can be usually carried out at the end point of the experiment. As an alternative, we have characterized the cytocompatibility of whole process, including encapsulation, hydrogel fabrication, and perfusion culture, as shown in Fig. 5.

Answer to the comment 3:

We agree that the quantitative characterization of cell viability after perfusion cell culture is very important. In order to characterize cell viability staining of living cells necessary in addition to the staining of dead cells shown in Fig. 5. Actually, we have tried staining of living cells after the perfusion culture (Fig. R1, below). However, the cells only in the area closed to the perfusion microchannel were stained probably due to the limitation in diffusion and metabolism of dye by living cells. Generally, dye molecule for living cell staining is permeable to cell membrane and converted to fluorescent and non-permeable molecule after hydrolysis in living cells. In addition, the conversion of dye molecule generally takes place less than 30 min in accordance to the protocol provided by the company. Due to these properties, dye molecules were consumed in the area closed to the perfusion microchannels and it is difficult to stain the living cells far away from the perfusion microchannel even though cells are alive. As a result, the cells in the center area of the 1,400 and 2,200 μm hydrogel region, which were approximately 300 μm far away from the perfusion microchannel, were not stained by dyes after 4 hour incubation (green in Fig. R1 below), even though not all of cells were dead in this region (red cells in Fig. 5), indicating that not all of cells were dead in the hydrogels with 1,400 and 2,200 spacing. Therefore, we think it is difficult to characterize cell viability properly in the center area of the 1,400 and 2,200 μm hydrogel region.

Due to above mentioned reason, we could not add the result on characterization of viability, even though we agree with the referee's comment on its importance.

Fig. R1 Trial for staining of living cells in the photodegradable hydrogel.

Answer to the comments 4 and 5:

In this paper, we report new method to generate dynamic concentration gradients. We think our method is potentially advantageous in studies in various biology fields. However, we are not intending to show application to specific biological event. Therefore, we focused on the demonstration of the newly established methods and the characterization of concentration gradients generated by the new method. As reviewer commented, we have carried out minimal biological experiment (live/dead staining) to demonstrate the applicability to biological experiment. We think our new method has unique advantage and applicable to various biological experiment because of its versatility. Therefore, our work can potentially contribute the advancement of the field of microfluidics even without the application to the specific biological event. We expect follow-up papers citing our papers will demonstrate the usefulness of our methods in specific applications such as embryology, immunology, angiogenesis, and others.

Referee: 3

Referee's Comments:

The authors conducted well designed investigation on the effect of the concentration gradient on the diffusive flux on encapsulated cells in micro-patterned hydrogel. In fact they succeeded to mimic the conditions in a real tissue. Of course this is not a scientific breakthrough as far as the first Fick's law is well known, but the contribution of the paper consists in building up of scaffold in which a real cells can be encapsulated and perfused by real patterned micro-channel. This opens a gate for many more detailed investigations of cultures of cells. I really like the work and recommend its publication.

Our response:

We thank reviewer's careful reading, understanding on our work, and evaluation for publication. We revised manuscript in accordance with the other referee's comments. We hope our manuscript is acceptable for publication in Royal Society Open Science.